# Multilevel Governance and Control of the COVID-19 Pandemic in the Democratic Republic of Congo: Learning from the Four First Waves

**DOI:** 10.3390/ijerph20031980

**Published:** 2023-01-20

**Authors:** Yves Kashiya, Joel Ekofo, Chrispin Kabanga, Irene Agyepong, Wim Van Damme, Sara Van Belle, Fidele Mukinda, Faustin Chenge

**Affiliations:** 1Centre de Connaissances en Santé en République Démocratique du Congo, Kinshasa 3088, Democratic Republic of the Congo; 2Kinshasa School of Public Health, University of Kinshasa, Kinshasa 127, Democratic Republic of the Congo; 3Public Health Faculty, Ghana College of Physicians and Surgeons, Accra MB 429, Ghana; 4Institute of Tropical Medicine, 2000 Antwerp, Belgium; 5School of Public Health, University of Western Cape, Cape Town 7530, South Africa; 6Faculty of Medicine, University of Lubumbashi, Lubumbashi 1825, Democratic Republic of the Congo; 7School of Public Health, University of Lubumbashi, Lubumbashi 1825, Democratic Republic of the Congo

**Keywords:** COVID-19, governance, decisions, response system, interventions, Democratic Republic of Congo

## Abstract

The COVID-19 pandemic continues to impose a heavy burden on people around the world. The Democratic Republic of the Congo (DRC) has also been affected. The objective of this study was to explore national policy responses to the COVID-19 pandemic in the DRC and drivers of the response, and to generate lessons for strengthening health systems’ resilience and public health capacity to respond to health security threats. This was a case study with data collected through a literature review and in-depth interviews with key informants. Data analysis was carried out manually using thematic content analysis translated into a logical and descriptive summary of the results. The management of the response to the COVID-19 pandemic reflected multilevel governance. It implied a centralized command and a decentralized implementation. The centralized command at the national level mostly involved state actors organized into ad hoc structures. The decentralized implementation involved state actors at the provincial and peripheral level including two other ad hoc structures. Non-state actors were involved at both levels. These ad hoc structures had problems coordinating the transmission of information to the public as they were operating outside the normative framework of the health system. Conclusions: Lessons that can be learned from this study include the strategic organisation of the response inspired by previous experiences with epidemics; the need to decentralize decision-making power to anticipate or respond quickly and adequately to a threat such as the COVID-19 pandemic; and measures decided, taken, or adapted according to the epidemiological evolution (cases and deaths) of the epidemic and its effects on the socio-economic situation of the population. Other countries can benefit from the DRC experience by adapting it to their own context.

## 1. Introduction

The COVID-19 pandemic has been costly and continues to impose a heavy burden on people around the world. Daily life has been profoundly altered, economies have entered into recessions, and many of the traditional social, economic, and public health safety nets that many people rely on in times of hardship have been put to the test [1]. On 30 January 2020, the World Health Organisation (WHO) declared COVID-19 to be a public health emergency of international concern, urging all countries to establish specific health policies to address this new public health problem [2]. The WHO declared COVID-19 a pandemic on 11 March 2020. This declaration promotes multi-level governance for each country to put in place strategies that will result in a network linking the various international and transnational institutions [3]. This collaboration should allow for the internalization of the benefits while adapting the measures or proposals according to the context of the country and the local circumstances [4,5]. Multilevel governance respond to the “Who Question” and describe the dispersion of authority whether this is within a state or beyond it, with drivers being ethno-territorial identity, democracy, interdependence, affluence, and peace [6,7,8].

The first cases of COVID-19 in Africa were recorded in Egypt on 14 February 2020 [9] and Nigeria on 27 February 2020 [10]. The pandemic has raised serious concerns about the ability of African countries to control the disease. The number of cases reported in Africa was not an accurate representation of the situation because of low detection capacity. African health systems were challenged with limited resources to control the pandemic; therefore, Africa needed to establish concrete strategies to deal with the disease [11,12]. This is why the African Union (AU) launched a new partnership program through the Centers for Disease Control and Prevention (CDC Africa) to accelerate the production and distribution of COVID-19 tests. More than one million tests were expected to be distributed on the African continent for screening [13].

The population of the Democratic Republic of Congo (DRC) in 2019 was 95.7 million over 26 provinces [14]. It is a country with a high burden of communicable diseases, with several epidemics such as Ebola, measles, cholera, and so forth [15,16]. This challenge is compounded by problems of geographical accessibility, the socioeconomic situation of the DRC, and outbreaks of armed conflict in some regions of the country.

The DRC recorded its first case of COVID-19 on 10 March 2020, and the cumulative number of COVID-19 cases was 87,023, with 1337 deaths in the period from 10 March 2020 to 8 May 2021 [17]. The epidemiologic situation was different according to the geographical position of the provinces with different gateways (air, land, and river). Therefore, the provinces also held autonomy, being a decisive criterion to ensure the execution of certain measures at the sub-national level as a response against COID-19. This means, according to Friedrich, that the legitimate decision of one level cannot be reversed by other levels without triggering a political, institutional, or even constitutional crisis [4].

The DRC’s health system is based on primary health care. It comprise three levels of health governance (central, provincial, and operational). Based on a theory of the state’s capacity to manage disasters, it is noted that the multiparty coordination of government for the delivery of public services delimits the dimensions of information capacity, coercion, design, and Execution of Decisions and Mobilization [5]. This may lead to externalities in the event of the response against an epidemic that affects several geographical and political entities. [18]. There are also two sectors involved in the health system: a public sector run by the government and a private sector with some facilities integrated into the national health system. This had implications for decision making in the national response to COVID-19, requiring a mix of strategies from both the national and decentralized entities to achieve an effective response [7,8]. Two research questions guided this study: (1) Why and how was the response to the COVID-19 pandemic organized at all levels of the DRC health system? and (2) How and what decisions were made to control this pandemic in the DRC?

The overall objective of this study was to explore national policy responses to the COVID-19 pandemic in the DRC and drivers of the response, and to generate lessons for strengthening health systems’ resilience and public health capacity to respond to health security threats.

## 2. Materials and Methods

Study site: This study was conducted in three provinces most affected by COVID-19 (Kinshasa, Haut Katanga and Nord Kivu) [19].

Study design: This was a single case study that took on a qualitative approach and literature reviews in deriving information about the national response to COVID-19 in the DRC.

Data collection: Data sources for this study consisted of a combination of document review, in-depth interviews, and non-participant observation during the first 24 months of the pandemic (March 2020 to March 2022). Then, each section was briefly described separately.

Literature review: The gray literature included national and international media reports on COVID-19, government and nongovernment documents developed to respond, and surveys conducted in relation to COVID-19 during the study period. This period was characterized by the occurrence of four waves, i.e., periods of increased COVID-19-related cases and deaths in the DRC.

The literature was searched on free search engines (Google, Google Scholar, PubMed) using keywords such as “COVID-19”, “decision”, “response system”, “interventions”, “communication”, “effects”, and “Democratic Republic of Congo (DRC)”.

The inclusion criteria for the study were any documents reporting on decisions made regarding the health of the Congolese population during the four waves of the COVID-19 pandemic, the organisation of the response to COVID-19, and the interventions that were implemented. All articles in French or English that met these inclusion criteria were included in the review. Any paper published in a language other than French or English was excluded. For gray literature, any empirical paper from an unidentified source was also excluded. Figure 1 summarizes the process of the identification and selection of the documents that were reviewed.

Each selected document was analyzed by three researchers (YK, CK, and JE) to extract the content in relation to each of the explored themes. Afterward, the three researchers met to pool convergent ideas, and divergent ideas were submitted for arbitration with a fourth researcher (FC).

Of the 508 gray literature documents and 86 scientific articles that were identified in the initial search, 46 gray literature documents (media articles, reports, and so forth) and 18 scientific articles that met the inclusion criteria were analyzed for three thematic groups related to the organisation of the response, the decisions of state authorities and the interventions that were implemented.

One out of four documents contained information on both the decisions made by the Congolese authorities and the interventions that were implemented to respond to the COVID-19 pandemic. One out of three documents described the multilevel organisation of the response to COVID-19 in the DRC (Table 1).

Key informant interviews (K): The in-depth interviews with key informants were performed to supplement the information from the document review to better understand the “why and/or how”, the obstacles/possibilities, the challenges of implementation, and the particular conditions of the context (localized interventions). In parallel with the decentralization of health competencies, key informants were purposefully selected, targeting actors involved in the response to the pandemic at different levels of the DRC health system. These were three central or macrolevel managers working at the General Direction of Disease Control (GDDC), the Division of Epidemiological Surveillance (DES), and the Technical Secretariat (TS). All of these actors were involved in the day-to-day management of the COVID-19 pandemic in the DRC. At the provincial or meso level, the heads of the provincial health divisions (PHDs) of the three most affected provinces (Kinshasa, Haut-Katanga, and North Kivu) were interviewed. These heads of the provincial divisions also act as incident managers for the COVID-19 epidemic in their respective provinces. At the operational level, six health zone managers (HZMs), including 4 in Kinshasa, 1 in Haut-Katanga, and 1 in North Kivu, were interviewed to document the approach to the response at the zonal level. A total of 12 key informants were interviewed.

A standard interview guide was used, but according to the specificity of the answers, probing questions were asked according to the hierarchical level of the key informant’s health pyramid, but also to the context of the region in order to bring out the reality on the field. These interviews were conducted by the research team in the provinces mentioned above. These face-to-face interviews were conducted from November 2020 to March 2021. Most of the interviews took place in the interviewee office and lasted 45 min on average.

Non-participant observation: Direct and indirect non-participant observation was used throughout the period of collecting data from decision makers and the population concerning the intervention implemented to the COVID-19 response. These observations made by the four authors who lived in the DRC and analyzed the data enabled them to qualify and even triangulate some of the information that was obtained from the interviews with the key informants.

Data management and analysis: The different documents selected for the study were centralized, examined, and then synthesized by identifying the different themes around the overall. All interviews were audio recorded and transcribed verbatim. Thematic analysis was done manually, which allowed for exploration and synthesis of codes, and themes and categories according to the study objectives. Data from key informant interviews and literature reviews were summarized and presented descriptively into the following themes: organisation of the response, decisions by state authorities, and interventions implemented. Similarities and contrasts were explored by drawing comparisons among the three groups of key informants to identify divergences and convergences in views.

## 3. Results

The results are presented according to the three main themes: the organisation of the response, the decisions of the state authorities, and the interventions that were implemented.

### 3.1. Organisation of the Response to COVID-19

Multilevel governance for the COVID-19 response was established at national, provincial, and operational (health zone and community) levels with multisectoral stakeholder representation, as summarized in Table 2.

At the national level, the organisational governance for the response included the Presidential Task Force, the Multisectoral Response Committee, and the Technical Secretariat [12,15,20,21,22]. The Technical Secretariat (TS) was the main hub of the response in the DRC. It was composed of representatives from the Ministry of Health, technical and financial partners, and civil society. The DRC, through this structure, developed a preparedness plan for the response in March 2020 to contain the spread of the virus [23]. The objectives of this plan were as follows: strengthen governance and prepare health zones; organize care structures for patients suffering from COVID-19; strengthen surveillance and case investigation capacities; improve infection prevention and control (IPC) and access to water, sanitation, and hygiene (WASH) services in all health structures and communities; strengthen risk communication and community engagement; and implement risk mitigation and social distancing measures. Based on the evolution of the epidemiological situation, the Technical Secretariat, through its scientific experts, proposed different scenarios to the political-administrative authorities, who then chose an option to implement to respond to the epidemic at the national level.

These different structures did not exist before the COVID-19 outbreak. Although they use Congolese state agents, they are ad hoc structures that operate outside the normative framework of the management of the Congolese health system.


*“The Technical Secretariat is the technical body of the multisectoral committee for information sharing and decision-making. It meets weekly and every Monday. There are also periodic meetings of coordination mechanisms between the Technical Secretariat, the Multisectoral Committee and the task force of the President to enable major decisions to be made, such as the application and release of curfews or the provision of strategic data to conduct the response.”*
(KI, GDDC)

As a result, the mechanism of coordination in the chain of information transmission among these different structures was not straightforward. They were operating outside the normative framework of the health system [24].


*“So, in terms of governance, it is a strategy that needs to be reviewed, especially in the mechanisms of collaboration among the different components. So, in a very complicated context, sometimes we do not know how to identify the limits of each other’s prerogatives and that can contribute to a blockage.”*
(KVIII, TS)


*“It was very difficult because at the beginning, we were left out of the management of the epidemic. They said it is the Technical Secretariat, it is the taskforce, it is... I do not know, it is happening at that level. They met. It is difficult at my level to tell you if they talk about what, what is the chain of transmission for decisions.”*
(KII, DES)

At the provincial level, a structure called the Provincial Coordination Committee (PCC), headed by the Provincial Governor and assisted by the Provincial Minister of Health and the Head of the Provincial Health Division (PHD), was set up. Other provincial ministries that were directly involved in the response, such as the Ministry of Communication, Planning, Education, and others, were also members of this committee. Some associations and nongovernmental organisations, such as technical and financial partners, were also included. The mission of this structure was to ensure the implementation of the interventions that were decided on at the level of the TS, the setting up of the different response commissions, and the application of risk mitigation and social distancing measures [21]. This structure was created to transmit the directives from the Technical Secretariat to the provinces and to manage the COVID-19 epidemic on a daily basis.


*“At the provincial level, the advantage is that it is the provincial Minister of Health comes to chair these meetings and we are with all these partners who support us; we discuss with them. If there are options to be considered, we do it together with them, and if there is information to share, we share it together. At the level of our coordination team, almost permanently, irregularly, we are with colleagues from communication, education, the Ministry of Planning, and Nonstate actors. We have a number of organisations that support the health system.”*
(KIII, PHD 1)

At the operational level, the Health Zone Management Team was in charge of implementing interventions in collaboration with the local political and administrative authorities. This included setting up rapid response teams, raising public awareness, training teams of providers, pre-positioning emergency kits, and managing suspected or confirmed cases (care and support, follow-up of contacts, lockdown/quarantine/isolation measures, dignified and safe burials).


*“We were sitting at the zonal level. We had local district committee meetings chaired by the municipality mayor. We had weekly meetings, and decisions were made during these meetings. When we meet, we see the priorities in relation to the activities to be carried out, we decide, we draw up a list of the decisions made, then we follow up under the aegis of the municipality mayor.”*
(KIV, HZM 1)

The DRC collaborated with many non-state partners, such as bilateral and multilateral development partners, international and local nongovernmental and civil society organisations, private companies, and religious organisations, in support of the response to COVID-19 in different sectors. Several activities were carried out together with these partners, including the prevention of COVID-19, the provision of water to improve WASH measures, the provision of IPC materials, and raising community awareness. Some partners actively participated in the care of patients in COVID-19 treatment centers (CTC) by providing materials and human resources for the medical and psychosocial treatment of COVID-19 patients. Technical and financial support was also provided to the various structures set up for the response to COVID-19 [25,26,27,28,29,30,31,32,33].


*“... Nonstate actors, we have a number of organisations that support the health system; in any case they are there, the list is so long and it depends on the level. However, at the level of the PHD we have, I can mention UNICEF, the WHO, FHI 360, MSF, and so forth. In any case, there are many partners who come to support the provincial health division.”*
(KIII, PHD1)

At the beginning of the COVID-19 epidemic, the government response was vertical with parallel structures. For example, there was a central response team that worked directly in the community and a provincial response team that worked directly with the community and was passively involved in the health zones, as the zonal chief medical officers did not have a monopoly on the response in their health zones. However, as the epidemic evolved, a zonal approach was adopted to decentralize management. Interventions were organized from the operational level, with the community, the health centers, and the reference hospital accompanied by the provincial health division. This verticalization of the control could be because the COVID-19 epidemic was a new phenomenon with a particular control approach that should be controlled by a few actors.


*“.... It is only with time that we finally thought of a kind of decentralization where the trainings were organized. We started to train the health zone management teams (HZMTs), which is the way the members of the HZMTs were trained. Then, we started to train the other providers (Titular Nurses and other professional categories). In addition, at our level of the zone, with the support of other partners such as MSF, PATH, and so forth, we started to train the community health workers (approximately 30 people). We trained the community relay agents (approximately 200), so that they in turn could pass on the information to the community so that it could adhere to the observance of the preventive measures (barrier gestures).”*
(KVI, HZM 5)

### 3.2. Decisions at the Central and Provincial Levels (Responses) and the Analytical Process

Public policies are essentially authoritative decisions made for society. Multiple decisions have been made for the control and management of the COVID-19 epidemic in the DRC since the first case was reported on 10 March 2020 [21]. Over time, some of these decisions have been strengthened, whereas others have been mitigated or abandoned altogether depending on the epidemiological evolution of the waves or periodic outbreaks of reported infections and deaths (Figure 2) and the observed effects of related interventions on the socioeconomic situation of the population.

Following the first resurgence of COVID-19, a state of health emergency was declared on 24 March 2020 by the President of the Republic for a period of 30 days in accordance with the Congolese constitution (Box 1). It was renewed for 15 days on several occasions up to 22 July 2020 [12,34,35,36,37,38,39]. On this occasion, several other measures were taken to reduce the spread of the disease in the country.


*“There were also major measures such as the closure of all the country’s borders with the outside world; over time, we wanted to confine the whole city, but it was not possible, and we ended up confining the commune of Gombe alone.”*
(KVII, PHD3)

Faced with the spread of the coronavirus, the governor of the city-province of Kinshasa announced an unpopular measure on 21 March 2020, namely, ‘In the central market as well as in the other markets, especially Marché de la Liberté, Marché de Gambela, Marché de Matete and all the other markets in the city of Kinshasa, only foodstuffs will be sold. The sale of other products is temporarily suspended’. This measure was not well received by the sellers in these markets, as a trade unionist reveals in these words:


*“We truly have irresponsible authorities who do not measure the consequences of their decisions.”*
[39]

The lockdown of the Congolese population, decreed on 28 March 2020, in the capital, which was the epicenter of the epidemic, was among the first nonpharmacological measures taken by the country’s authorities. However, this decision was revised on 2 April 2020, declaring a partial lockdown of Gombe, the hotspot of the epidemic in Kinshasa [12,40,41]. This measure gave rise to many protests caused by not only the lack of resilience of the population but also the lack of preparation of the government and the lack of consequent accompanying measures to guarantee the population’s food security. Indeed, even among the ranks of the police, soldiers and civil servants, the total lockdown was not accepted, and this can be explained by two main reasons. First, it should be noted that their wives and children are among the small-scale operators in the informal sector, and the income earned is an important source of income for the households concerned. Second, a total lockdown means that the same government officials can no longer access the informal taxes that they used to collect from informal operators (sellers, traders, drivers, and so forth) [39]. As a result, the option of a partial lockdown for only the most affected municipality was considered.

A key informant said, *“For example, we refused to do a global lockdown; that was a decision that had been made, and then we had to postpone it, and that is why we had only contained Gombe. This means that now, we do not do lockdowns like we did in Gombe anymore because we had feedback; we changed the lockdown to a curfew, and that was by having feedback from the community.”*(KVIII, TS)

In the province of Haut-Katanga, the governor of the province decreed a total lockdown for 48 h following the detection of two suspected cases from the capital of Kinshasa on 22 March 2020 [42]. Another short-term lockdown was also carried out in the province of North Kivu on 6 April 2020, with the isolation of three towns, namely, Goma, Beni, and Butembo.

To mitigate the financial expenses for the population caused by nonpharmacological measures, on 20 April 2020, the government decided to suspend the collection of value-added tax (VAT) on certain basic necessities, suspend payments of electricity and water bills, and issue a moratorium for people who could not pay house rent for a period of three months. This decision was made to remedy the negative impact of COVID-19 on the economy, including the volatility of prices for basic necessities and the mobilization of resources by the population [43]. This situation was experienced by most Congolese. As a result, the informal sector was in jeopardy, as 55% of households saw their incomes fall, and 1/4 of households overall and 40% of low-income households were unable to cover their expenditures on food, housing, transportation, and so forth. Almost the majority of the households in the DRC reduced their consumption of goods and services during the pandemic due to a lack of income [44].

The change in alert level was linked to the screening of cases across provinces to improve the capacity of the provincial health system to control the situation (Interview 4).

A key informant said: *“Well, at the level of the city of Kinshasa, given that at the first moment (March, April, May), we realized that the city of Kinshasa (the capital) was recording a greater number of cases and the municipality of Gombe recorded more cases at the beginning of the pandemic. Decisions were made, notably for lockdowns, which essentially concerned the city of Kinshasa in general, and especially the municipality of Gombe, where there were no more entries or exits and other measures were taken: the banning of gatherings, the closures of schools and churches, the closures of all cultural, commercial and sporting activities, and so forth. In short, all mass activities were banned. These are the measures that were taken.”*(KI, GDDC)

The number of cases increased until they peaked in June and then began to decrease. The decrease in cases led to the release of the lockdown in the municipality of Gombe [45]. On 21 July 2020, the President of the Republic announced that the procedure be followed for the gradual resumption, from 22 July to 15 August 2020, of the activities suspended under the state of health emergency, which had been enforced since 24 March 2020 [43,44,45].

In Kinshasa, by September 2020, the population had relaxed barrier measures. However, failure to comply with these measures exposed more than one person to COVID-19 [46,47]. This behavior was especially noticeable in places of mourning and churches, at festivals and even in some institutions [48]. Following a rise in COVID-19 cases in November 2020, the authorities, in collaboration with the Multisectoral Committee, renewed certain measures that were taken during the first wave of the COVID-19 pandemic. During the month of December 2020, a curfew was introduced, accompanied by the following: the application of compulsory tests for internal travelers and those coming from and going abroad; a renewed ban on festive ceremonies and meetings of more than 10 people; the closure of discotheques and nightclubs; the postponement of the resumption of academic activities at the level of higher education; early holidays for primary and secondary school pupils; the compulsory wearing of masks; the respect of physical distancing; the systematic washing of hands; the taking of temperatures; and the prohibition of public marches, artistic productions, and kermesses [49,50]. Non-compliance with the mask requirement was punishable by a fine of approximately USD 2.5. This sum was collected by the police to encourage the population to strictly respect the barrier measures [51]. However, this measure was not properly applied because the police offers shared the funds they collected among themselves and with their hierarchical superiors [39]. All the measures (Table 3), with the exception of the curfew and vaccination measures, were taken starting with the first wave; these measures were essentially imposed by the international evolution of the pandemic and on the basis of the contextual and epidemiological analyses of the response that were provided by the Technical Secretariat, which was headed by an eminent scientist of international renown with substantial experience managing epidemics of emerging viral diseases such as the Ebola virus, which is raging in the DRC on a recurrent basis. Certain measures have been permanently and continuously maintained during and between waves. These included the measures whose acceptance and practice were relatively less contested by the population.

Other measures were taken only during the first wave or intermittently between waves. These included lockdowns, the closure of schools and universities, the closure of discotheques and drinking establishments, the prohibition of the organisation of festive ceremonies, and so forth. These measures were imposed upon those whose socioeconomic impacts on the population were very significant and were strongly contested by the population. They were thus taken to reinforce the permanent measures to effectively deal with epidemic outbreaks that heralded the start of a wave.

A series of other measures (the suspension of value-added tax, the suspension of water and electricity bill payments, the postponement of rent payments for tenants) were taken, essentially during the first wave, to mitigate the socioeconomic impact of certain other measures on the population. Naturally, these measures were well appreciated by the population, which also considered their duration relatively short.

### 3.3. Interventions at the Operational Level

At the health care facility and community levels, the response to COVID-19 included a number of components related to patient management, case detection, surveillance and investigation, infection prevention and control (IPC/WASH), communication, and community engagement.

For the management of patients with COVID-19, the grouping of health zones around one or two health care facilities, the development of targeted health facilities, the clinical classification of cases, and the implementation of triage units were performed. The treatment of moderate and severe COVID-19 patients included hospitalization in COVID-19 treatment centers (CTCs) for supportive care, with oxygen and anti-coagulation according to WHO guidelines. In practice, care providers at COVID-19 treatment centers routinely followed a regimen using products such as azithromycin, zinc, vitamin C, paracetamol, and hydroxy chloroquine [21,28,30,52].


*“... For the management of COVID-19 patients, there were structures that were identified here in Kinshasa and that we gave a lot of materials and equipment in terms of COVID-19 treatment centers in hospitals, and we trained the staff of these hospitals to be able to take care of simple cases, moderate cases and severe cases of COVID-19. We also had to train the teams in the health zones because at a certain point we saw that the hospitals were saturated, so we had to rely on the structures in the health zones to take care of patients in the community. Not all patients should always go to hospitals.”*
(KI, GDDC)

For laboratory diagnosis, the following strategies were implemented: strengthen health personnel for COVID-19 diagnosis; support the functioning of the National Institute of Biomedical Research (NIBR); extend diagnoses to other laboratories; perform household/high-risk contact screening around the confirmed case; mass screening in hotspots such as the one carried out at the Martyrs’ Stadium in Kinshasa on 26 May 2020; strengthen the information system in the laboratory network; and strengthen biosafety/biosecurity and quality control in the targeted laboratories [20,27,53].

For surveillance and case investigation, the following were performed: building the capacity of providers; strengthening passive surveillance at the health zone level; setting up isolation areas and rapid response teams; actively finding cases; and strengthening health information management [23,28] (Table 4).


*“.... after receiving this alert, they carry out investigations in the field, that is to say, correctly identifying the person and collecting the signs that he or she presents in relation to a table of signs that we provided that are consistent with COVID-19. Thus, after having collected the signs, interviewed or talked to the patient, when we find that this patient needs to be collected, i.e., when the alert is validated, the person goes to the lab to collect the sample and from there, the sample is taken to the NIBR...”*
(KIX, HZM 3)

For IPC/WASH, the strategy was to build the capacity of clinical and community providers, provide IPC/WASH tools, guidelines and equipment, evaluative supervision and strengthen waste management [20,32,54].


*“... There is also an intervention through the IPC (Infection Prevention and Control) subcommittee, which is in charge of household decontamination. If a patient is already positive and has already been treated, they quarantine at home, and the house must be decontaminated to protect the family members with whom the patient lives so that they cannot catch the disease. Even in the offices where we work, if there is already a positive case, in any case, we have to close the offices and decontaminate them.”*
(KX, HZM 2)

Risk communication and community engagement involved developing key messages, managing rumors, producing and disseminating communication tools, raising community awareness, and creating a green line. As a result, radio and television broadcasts were organized, informative SMS messages were sent to mobile phones, posters were placed in strategic locations in different cities, and leaflets were distributed [15,53,54,55,56,57,58,59].

As one key informant said, *“Communication in general is done through sensitization by the community facilitators and community relay agents. That is, there is a program in place that each community relay agent follows, as there are many of them; those who have been trained by the zone, they share the avenues or health areas to cover. In addition, every day, they go to the places where there are a lot of people, places that are more frequented, for example, markets, shops, and avenues; they do the sensitization with megaphones and they insist a lot in the hot spots where there are crowds of people. All they say is to observe the barrier gestures.”*(KX, HZM 2)

Epidemiological data were used to determine the most at-risk areas for implementing maximum prevention and control measures. These interventions were adapted to the socioeconomic and cultural levels of the communities. These strategies were drawn from the substantial experience of managing epidemics, especially the Ebola virus.

On 19 April 2021, the DRC launched the COVID-19 vaccination campaign. The campaign continued in the first five active provinces that were selected, namely, Haut-Katanga, Kongo Central, Lualaba, North Kivu, and South Kivu. Nearly 1.7 million doses of the Oxford-AstraZeneca vaccine, manufactured in India, were delivered to the country. The batch of vaccines arrived from Mumbai on 2 March 2021, via the COVAX mechanism. Vaccination was voluntary for health workers and other vulnerable groups, such as people above 55 years of age as well as anyone with comorbidities (diabetes, high blood pressure, obesity or any other chronic disease) [58] (Table 4).

The statement of a key informant is as follows: *“...We have seen in the past that the introduction of vaccines related to other diseases was beneficial. In addition, we even think that this time, this vaccine could be the solution to this pandemic, especially since we have even fewer cases in the country and the case fatality rate in relation to the pandemic or disease is even lower. So it is time. It is a decision that allows us to anticipate the danger...”*(KIX, HZM 3)

The vaccination campaign was not well received by the Congolese population. During the first few days, more than 80% of the people vaccinated in the different hospitals were of foreign nationality (Indian, Lebanese, Belgian, Greek, and so forth) [59]. Despite the multiplication of vaccination sites in the various provinces that were most affected, the cumulative number of people vaccinated by 30 June 2021 was only 59,244 [60]. For a country of nearly 100 million people, the number of people vaccinated with the first batch of the AstraZeneca vaccine was insignificant [14]. Certainly, there was a reluctance of the population to accept the AstraZeneca vaccine, as was the case in other African and Western countries, because of media reports that raised doubts about the safety of the vaccine, for example, that it would cause blood clots in vaccinated people. However, the risks are extremely limited. This mistrust was further accentuated by the reluctance of state officials. Although the Minister of Public Health had received the vaccine with other officials, the President of the Republic had not been vaccinated and had clearly expressed his reluctance in a radio interview.

A key informant said, *“As always! Even the disease itself has not been easily accepted, let alone the vaccine. To date, if you just look at the numbers of people vaccinated, it is not truly too great. Since we started the COVID-19 vaccination activity on May 12 in our zone, we have not yet reached 20 people vaccinated.”*(KX, HZM 2)

## 4. Discussion

In this study, the collected data provided a clear picture of the response to the COVID-19 pandemic in the DRC. Recent experiences with epidemics of Ebola, measles, and so forth, were capitalized on, including coordination, laboratories and research, case management, infection prevention and control, community engagement, and logistics and surveillance, particularly in the follow-up of contacts [55].

In the organisation of the response to COVID-19, the DRC developed several strategies to strengthen the governance of actors at each level of the health pyramid. As a result, multi-level governance structures were established, such as the task force composed of advisers to the President of the Republic who reported directly to the President, the Multisectoral Committee that reported to the Prime Minister, who was supported by the Minister of Health, and the Technical Secretariat that reported to a coordinator appointed by the President of the Republic. The establishment of a technical secretariat was inspired by recent experience with the Ebola epidemic. At the beginning of the pandemic, the response was centralized at the top through the Technical Secretariat without direct involvement of the provincial and operational teams, especially since the city of Kinshasa (the capital of the DRC) was the epicenter. Such a policy of centralization of power and decision-making in the response to the first wave was also originally considered in other countries including Guinea [61]. However, with time and the evolution of the epidemiological curve, the provincial coordination committees took over, making the health zone the operational unit of the response to COVID-19. This decentralization brought the actors closer to the community by also integrating the different stakeholders at the peripheral level. The main argument is that successful control over the pandemic depends on effective and integrated leadership at different levels [62,63]. As a result, the political and administrative authorities and the various associations took part in raising awareness among the population. Unlike the DRC, in Uganda, the governance of the response respected the normative framework of its health system. The impulse came directly from the Prime Minister, who was supported by the Ministry of Health, which had a national unit known as the “response force” responsible for carrying out field activities. This unit collaborated with the various structures at the district level to implement several strategies to respond to COVID-19, such as risk communication, increased awareness, social mobilization activities, and a community-based disease surveillance model [55]. However, in Nigeria, a multi -sector national coronavirus preparedness group (NCPG) has been set up for coherent and effective coordination of the country’s preparedness efforts [64]. Following the rapid spread of COVID-19, several countries, including the DRC, implemented various nonpharmacological measures, such as lockdowns, the closure of air and land borders, travel restrictions in different provinces, the closure of schools and universities, the closure of bars, the enforcement of social distancing with the prohibition of gatherings, curfews, and so forth [64,65].

Given the difficult context for the implementation of a full lockdown, the DRC implemented a partial lockdown in the municipality of Gombe, which had the most COVID-19 cases in Kinshasa, the epicenter of the disease at the time. However, in two other towns, a short-term total lockdown was carried out to allow for rapid interventions and investigations in the field. However, during the second outbreak, this measure was abandoned in favor of a curfew. A few African countries were able to implement lockdowns, such as those in the Accra and Kumasi metropolitan areas in Ghana and in several cities in Nigeria, ranging from two weeks to a month. However, in South Africa and Uganda, home lockdowns were implemented throughout the country, with a strict curfew that included a ban on all exercise outside the home. In addition, in Rwanda, the application of these containment measures was ensured by police officers regulating traffic within and between districts [66]. Some essential activities were allowed between 8 am and 5 pm. Other countries, such as Tanzania and Zambia, had not opted for a lockdown or curfew, but instead encouraged self-isolation for anyone suspected of having interacted with an infected person [63,67].

The ban on gatherings was a measure taken after the release of lockdown. However, this measure in the DRC evolved with the epidemiological situation, from 20 people to 100 people. This situation was also observed in other African countries, such as Botswana, where on 16 March 2020, gatherings of more than 100 people were banned, and four days later, gatherings of 10 or more people were also banned. Ghana suspended all public gatherings of two or more people, whereas Malawi and Sierra Leone allowed public gatherings of up to 100 people. In the DRC, funeral ceremonies were banned with the bodies being taken directly to the cemeteries. Only the immediate family was allowed to attend the removal of the body from the morgue at the beginning of the epidemic. Although authorities wanted to reduce crowds at the morgue, the high number of body removals, usually organized on weekends, often led to crowds at morgues in different towns in the DRC. The number of people allowed to attend funerals varied considerably between African countries; Kenya limited funeral attendees to immediate family members, Rwanda limited funerals to 10 people, and South Africa allowed up to 50 people. Although Ghana banned all gatherings of 2 or more people, an exception was made for private funerals with up to 25 participants. In Sierra Leone, gatherings at funerals were limited to 20 family members [68,69].

The Congolese authorities also made the decision to close schools and universities to limit gatherings. This decision was accompanied by certain measures, such as the organisation of courses by certain television channels to keep children occupied. Awareness was raised among parents to encourage children to work at home. Some public schools were more organized by using social networks for distance learning between teachers and pupils. Despite these various measures, the education sector was significantly affected, and thus far, school and university activities have not yet resumed at the usual pace following the repeated suspension of school and student activities. This situation was also observed in other African countries, such as South Africa, where the suspension of teaching and learning activities also affected learning and the academic calendar. However, the government, through the Department of Education, implemented strategies such as online support programs for learners, similar to the COVID-19 curriculum support programs for learners. Higher education and learning institutions adopted blended learning using a learning management system (Moodle), as well as social media, platforms such as WhatsApp and video conferencing platforms such as Zoom, Skype, and Microsoft Teams, among others [70].

In terms of operational interventions, the response to COVID-19 was very much inspired by the response to Ebola. The latter’s main strategy was to break the chain of transmission by ensuring rapid detection and the isolation of cases in specific treatment centers, intensifying multidisciplinary public health measures, increasing certainty regarding confirmed cases, strengthening communities, and providing engagement activities. Finally, vaccine trials were also used to prevent the spread of cases [55,71]. Regarding the response to COVID-19, the Congolese authorities opted to screen COVID-19 patients to ensure prompt treatment. Despite the benefits of large-scale screening for COVID-19 that have been demonstrated in several countries, the DRC did not have this capacity. As a result, screening was focused on suspected cases and travelers, who were required to be tested 48 h before flying to or from the DRC. The DRC was able to organize diagnostic centers for COVID-19 using RT–PCR and GeneXpert, which were already present in the country for the diagnosis of other diseases, such as TB and HIV. However, to increase the number of diagnostic sites, other laboratories were allowed to use rapid antigenic diagnostic tests. This situation was not specific to the DRC and was present in most countries, including the majority of African countries. These countries faced many challenges in their efforts to diagnose suspected cases, perform contact tracing for further testing, and monitor test deployment. A major diagnostic challenge arose from the nature of RT–PCR testing, which is the gold standard for COVID-19 testing. RT–PCR test kits are expensive, making large-scale testing cost-prohibitive. In addition, the test requires expensive equipment, including PCR machines and equipped BSL-2 laboratories. These factors and the need for trained personnel mean that in most African countries, very few centers have the capacity to test for COVID-19. In fact, RT–PCR diagnostic capacity is severely limited in most African countries, with the number of qualified testing laboratories ranging from 1 to 3 in 40 African countries [71]. However, this situation improved over time and during the evolution of the epidemic. In Rwanda, for any traveler, a negative COVID-19 PCR test had to be no later than 120 h before the flight [66].

With research, vaccines were produced between 2020 and 2021 to help respond to COVID-19. Thanks to the COVAX program, African countries have benefited from vaccines, including the DRC. The DRC launched the vaccination campaign in April 2021 with the AstraZeneca vaccine. This campaign was controversial, with its suspension occurring even before it was launched; this led to mistrust, even though at the outset, a small portion of the population was willing to be vaccinated [72,73]. During the study carried out by the European Union, which noted a few cases of thromboembolism, a decision was made to temporarily suspend the campaign to conduct further research. However, vaccination continued in the European Union a few weeks later with the AstraZeneca vaccine. Despite these results, more than three countries in Africa and the Middle East had negative attitudes toward COVID-19 vaccines [63]. South Africa had already suspended the use of the AstraZeneca COVID-19 vaccine in March 2021. This was because the vaccine was not able to stop the spread of the virus variant that was predominantly circulating in South Africa at that time [74]. However, the DRC was able to acquire several other vaccines that are currently being given to the population, including the Pfizer, Johnson-Johnson, Moderna, and Sinovac vaccines, to reduce the population’s distrust and increase the possibility for individuals to receive a vaccine of their choice. However, the COVID-19 vaccination campaign still seems to be in trouble.

## 5. Limitations of the Study

Two limitations can be noted for this study. First, the study mostly relays on online sources for the chronology of interventions implemented in the DRC; fortunately this has been confirmed by author’s non-participant observation. Second, in-depth interviews only conducted with state actors could have led to a one-sided view of the response, but this has been mitigated by considering all actors, including non-state actors in the strategic organisation of the response to COVID-19 in the DRC (Table 2).

## 6. Conclusions

This study aimed to explore national policy responses to the COVID-19 pandemic in the DRC during the four first waves and drivers of the response, and generate lessons for strengthening health systems resilience and public health capacity to respond to health security threats. Lessons that can be learned from this study include the strategic organisation of the response inspired by previous experiences with epidemics; the need to decentralize decision-making power to anticipate or respond quickly and adequately to a threat such as the COVID-19 pandemic; and measures decided, taken, or adapted according to the epidemiological evolution (cases and deaths) of the epidemic and its effects on the socio-economic situation of the population. However, there is still a need for increased community engagement in public health interventions, especially regarding the vaccination actually recognized as one of the effective strategies in COVID-19 response, given that the country may not be spared from another outbreak of COVID-19. Further studies that could be undertaken include the identification of the reasons of the weak community engagement in COVID-19 vaccination and how to address them, as well as the assessment of the cost-effectiveness of setting up ad hoc structures to respond to an emerging disease. Other countries can benefit from the DRC experience by adapting it to their own context.

## Figures and Tables

**Figure 1 ijerph-20-01980-f001:**
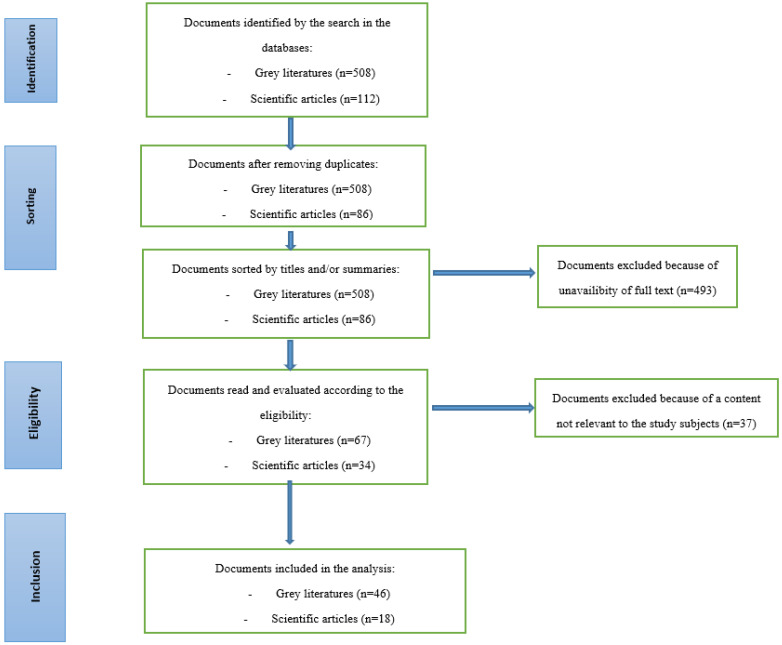
Document selection process (Prisma diagram).

**Figure 2 ijerph-20-01980-f002:**
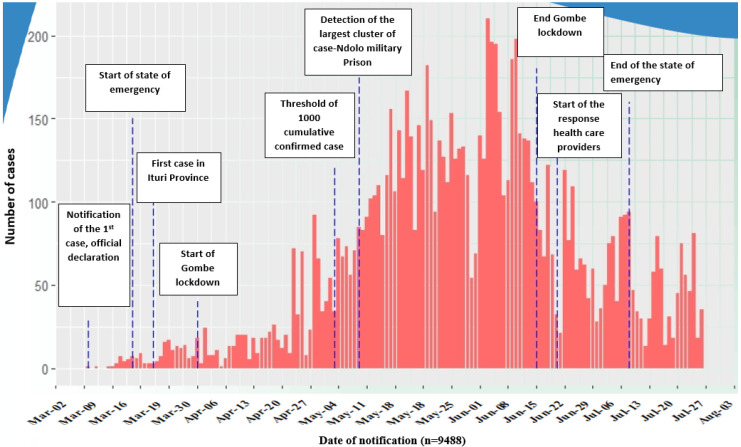
Evolution of political and health decisions in relation to the daily reporting of COVID-19 cases in the Democratic Republic of Congo from 10 March to 9 August 2020, and major events [28].

**Table 1 ijerph-20-01980-t001:** Distribution of documents analyzed by category and theme.

Category of Documents	Themes Used		
	Organisation of the Response	Decisions by State Authorities	Interventions Implemented
Gray literature documents	12	21	13
Scientific publications	3	4	11
Total	15	25	24

**Table 2 ijerph-20-01980-t002:** Strategic organisation of the response to COVID-19 in the DRC.

Structures	Composition	Role
State actors at the central level		
Presidential Task Force (PTF)	President of the Republic Advisors	✓ Acts as an interface between the Technical Secretariat and the Presidency
COVID-19 Multisectoral Response Committee (MRC)	✓ Prime Minister ✓ Minister of Health ✓ Other related ministers	✓ To provide the government’s policy direction ✓ Mobilizing resources and managing funds
Technical Secretariat (TS)	✓ Coordinator ✓ General Secretary of Health ✓ General Inspector of Health ✓ Representatives of technical and financial partners ✓ Civil society representatives	✓ Provide strategic management of all response and preparedness commissions.
State actors at the provincial and peripheral levels		
Provincial Coordination Committee (PCC)	✓ Governor ✓ Ministry (health, communication) ✓ Provincial Division Manager ✓ NGOs	✓ Implementation of interventions decided by the TS ✓ Coordination of the response at the provincial level
Operational structure	✓ Health Zone Management Team ✓ Local political and administrative authorities	✓ Establishment of rapid response teams ✓ Raising public awareness ✓ Decentralized management of the epidemic through a local approach
Non-state actors		
Technical and financial partners	✓ WHO, World Bank, Doctor Without Borders, UNICEF, Alima, ICRC…	✓ Technical support for the government ✓ Supply of materials ✓ Strengthening the capacity of providers ✓ Case management ✓ Raising public awareness
Other actors	✓ NGOs and NPOs ✓ Private companies ✓ Religious organisations	✓ Raising public awareness ✓ Supply of materials ✓ Strengthening the capacity of providers

**Table 3 ijerph-20-01980-t003:** Measures to prevent COVID-19 and mitigate its effects on the population in the DRC.

Taken Measures	Year 2020										Year 2021												2022			Acceptability
	M3	M4	M5	M6	M7	M8	M9	M10	M11	M12	M1	M2	M3	M4	M5	M6	M7	M8	M9	M10	M11	M12	M1	M2	M3	
	1st Wave								2nd Wave						3rd Wave							4th Wave				
Permanent measures																										
Correct wearing of mandatory masks	X	X	X	X	X	X	X	X	X	X	X	X	X	X	X	X	X	X	X	X	X	X	X	X	X	Very acceptable by 53% of people
Systematic hand washing	X	X	X	X	X	X	X	X	X	X	X	X	X	X	X	X	X	X	X	X	X	X	X	X	X	Very acceptable
Physical or social distancing	X	X	X	X	X	X	X	X	X	X	X	X	X	X	X	X	X	X	X	X	X	X	X	X	X	Weakly acceptable
Discouragement of greeting by hand when meeting	X	X	X	X	X	X	X	X	X	X	X	X	X	X	X	X	X	X	X	X	X	X	X	X	X	Moderately acceptable
Avoid touching the face (eyes, nose, mouth) without disinfecting the hands	X	X	X	X	X	X	X	X	X	X	X	X	X	X	X	X	X	X	X	X	X	X	X	X	X	Although very acceptable, impossible to achieve
Cough or sneeze into the elbow with a disposable tissue	X	X	X	X	X	X	X	X	X	X	X	X	X	X	X	X	X	X	X	X	X	X	X	X	X	Very acceptable by 53% of people
No hug as a greeting when meeting	X	X	X	X	X	X	X	X	X	X	X	X	X	X	X	X	X	X	X	X	X	X	X	X	X	Moderately acceptable
Border screening	X	X	X	X	X	X	X	X	X	X	X	X	X	X	X	X	X	X	X	X	X	X	X	X	X	Very acceptable
Intermittent measures																										
Prohibition of gatherings of more than 20 people	X	X	X	X	X	X					X	X	X	X												Moderately acceptable
Closure of schools	X	X	X	X	X	X					X	X	X	X												Acceptable by 37% of people
Closure of Universities and Higher Institutes	X	X	X	X	X	X					X	X	X	X												Acceptable by 37% of people
Suspension of services, funerals, and sports activities	X	X	X	X	X	X					X	X	X	X	X	X	X	X	X	X	X					Acceptable by 34.2% of people
Suspension of sports activities and non-essential shops	X	X	X	X	X	X									X	X	X									Unacceptable
Closing of discos and nightclubs	X	X	X	X	X	X									X	X	X									
One-time measures																										
Suspension of all flights from countries at risk	X	X	X	X	X	X																				Weakly acceptable
Border closure	X	X	X	X	X	X																				Weakly acceptable
Restriction of provincial movements	X	X	X	X	X	X	X	X																		Weakly acceptable
Prohibition of all migratory movements	X	X	X																							Weakly acceptable
Prohibition of river transport between provinces	X	X	X																							Weakly acceptable
Quarantine of travellers, suspected and positive cases	X	X	X	X	X	X	X	X																		Moderately acceptable
State of health emergency	X	X	X	X																						
Confinement	X	X	X	X	X																					Unacceptable by 80% of Congolese in Kinshasa
Closure of public establishments				X	X	X																				Unacceptable by 67% of people
Value Added Tax exemption for two months		X	X	X																						Very acceptable
Suspension of payment of electricity and water bills for two months		X	X	X																						Very acceptable
Postponement of the rent payment by tenants		X	X	X																						Very acceptable
Curfew from 9 p.m. to 5 a.m.									X	X	X	X	X	X	X	X	X	X	X	X	X					
Vaccination against COVID-19														X	X	X	X	X	X	X	X	X	X	X	X	Very acceptable

M = Month; X = Measure implementation time.

**Table 4 ijerph-20-01980-t004:** COVID-19 case detection and treatment measures implemented in the DRC.

Taken Measures	Year 2020										Year 2021											Year 2022			
	M3	M4	M5	M6	M7	M8	M9	M10	M11	M12	M1	M2	M3	M4	M5	M6	M7	M8	M9	M10	M11	M12	M1	M2	M3
	1st Wave								2nd Wave						3rd Wave							4th Wave			
Active case finding	X	X	X	X	X	X	X	X																	
Case detection (rapid test and PCR)	X	X	X	X	X	X	X	X	X	X	X	X	X	X	X	X	X	X	X	X	X	X	X	X	X
Mass screening			X	X	X																				
Vaccination against COVID-19														X	X	X	X	X	X	X	X	X	X	X	X

M = Month; X = Measure implementation time.

## Data Availability

The data used for this study are available from the corresponding author upon reasonable request.

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
