# Peer review of "Multilevel Governance and Control of the COVID-19 Pandemic in the Democratic Republic of Congo: Learning from the Four First Waves"

_ijerph, 2023, doi:10.3390/ijerph20031980_

Round 1

Reviewer 1 Report

This paper is a case single-case study, composed of various published literatures plus some interviews with state actors. The paper gathers some welcome and useful information on the DRC's dealing with Covid-19 pandemic, and provides, as the authors themselves suggest on line 521, 'a clear picture' of recent developments in the DRC. However, it does little more than that.

To begin with, there is no research question in the strict sense of the term, beyond as to 'what happened'. There is also no particular argument being made, no proper hypothesis being developed in light of the insights gained that could be taken up by other researchers in the field, and possibly beyond. More specifically, there is no theoretical contextualization of this 'report of events' whatsoever, and no linking to any of the literatures to which this study could have meaningfully be related to (such as the crisis leadership literature, or the multi-level governance literature more generally). Further, the authors also do not put their findings into systematic perspective by any comparative analysis (though I note there are a few sentences on other African countries in lines 556ff). Finally, the authors do offer any generalizable propositions when presenting their 'lessons learned', and fail to discuss the wider implications of their results for the social science theory, or public policy makers for that matter.

This reviewer believes that this is not enough for an original research article in a Web of Science-listed journal.

That apart, I am also a bit surprised to see that scholars who apparently provided some pre-submission advice (reading and commenting a previous draft) seek to claim a full co-authorship for this piece.

Author Response

Thank you for your comments that helped us to improve this scientific work. Please find our answers in the pdf document

Reviewer 2 Report

The introduction can be much improved. It is not very interesting. There are missing references and also in the conclusions. The article is interesting but can be improved. Congratulations.

Author Response

(The authors gave the same response as above.)

Reviewer 3 Report

Please check the comments in the PDF file.

Author Response

(The authors gave the same response as above.)

Round 2

Reviewer 1 Report

The authors have demonstrated their willingness to engage with this referee's earlier suggestions, and I am thus happy to recommend this revised draft for publication.